# Chlorine in an Organic Molecule, a Universal Promoter—Workhorse—Of Reactions

**DOI:** 10.3390/molecules28247957

**Published:** 2023-12-05

**Authors:** Mieczysław Mąkosza, Michał Fedoryński

**Affiliations:** 1Institute of Organic Chemistry, Polish Academy of Sciences, Kasprzaka 44/52, 01-224 Warsaw, Poland; 2Faculty of Chemistry, Warsaw University of Technology, Noakowskiego 3, 00-664 Warsaw, Poland

**Keywords:** phase transfer catalysis, γ-halocarbanions, dichlorocarbene, vicarious nucleophilic substitution, nucleophilic substitution of chlorine, base induced β-elimination, α-halocarbanions, Darzens reaction, chlorine as electrophilic center

## Abstract

Due to the electronic configuration of the atom and charge of the nucleus, the chlorine in organic molecules can exert a variety of effects. It can depart as a chloride anion in the process of substitution and elimination, facilitates the abstraction of protons and stabilizes generated carbanions, exerts moderate stabilizing effect of carbenes, carbocations and radicals. There are frequent cases where chlorine substituent promotes more than one transformation. These rich effects of chlorine substituent will be illustrated by examples of our work.

## 1. Introduction

From the beginning of organic chemistry, particularly organic synthesis, chlorine substituents were widely used to perform a variety of transformations. Early empirical observations were generalized with the progress of physics and allowed for the determination of the structure of atoms. The development of quantum mechanics made it possible to explain the unique properties of chlorine substituents in terms of modern orbital concepts. Chemists working in the field of organic synthesis use chlorine substituents to solve a variety of current problems and usually are not interested in the great variety of other possible reactions promoted by chlorine.

Therefore, we have decided to present a variety of transformations that can be performed thanks to the presence of chlorine in a molecule. We expect that presenting these in a rational way, as a consequence of the electronic structure of chlorine substituents, can inspire the readers to look for new reactions. In a few cases, we described reactions with bromo compounds. Bromine atoms can promote similar reactions; however, because of the lower energy of the bromine–carbon bond, larger size of the atom and its electronic configuration, it is less universal.

Due to the electronic configuration of the chlorine atom and the charge of the nucleus, the chlorine in organic molecules can exert a variety of effects. Due to the polarity and relatively moderate energy of the carbon–chlorine bond, it can depart from organic molecules as a chloride anion in the process of substitution and elimination. Due to electron-withdrawing effect and low laying 3d orbitals, chlorine facilitates the abstraction of a proton and stabilizes generated carbanions. It also exerts a moderate stabilizing effect of carbenes, carbocations and radicals.

The substitution by a nucleophile proceeds as a synchronous process according to the S_N_2 mechanism, or via initial dissociation of the carbon–chlorine bond, resulting in formation of chloride anion and carbocation, followed by coupling of the carbocation with a nucleophile, S_N_1. Furthermore, when chlorine is connected with the electron-deficient π-system as in the case of acid chlorides, chloronitroarenes, β-chlorovinyl ketones, etc., the addition of nucleophiles to the carbon atom connected with chlorine, followed by departure of chloride anion from the intermediate adducts, takes place.

Substitution of halogen atom, connected to aromatic ring, with highly basic nucleophiles can proceed via elimination-addition process with the aryne intermediacy.

Nucleophilic substitution reactions of halogen in aromatic haloarenes initiated by transfer of single electron to chloroarenes, followed by dissociation of the produced anion radical (S_RN_1), are also known.

The elimination can proceed as the spontaneous departure of the chloride anion from the negatively charged carbon—α-elimination to form carbenes, and three variants of β-elimination. The departure of the chloride anion occurs synchronously with the abstraction of a proton from vicinal carbon (E2), the spontaneous dissociation of the carbon–chlorine bond to form chloride anion and carbocation, subsequently losing a proton (E1) and the departure of the chloride anion promoted by the generation of carbanion on β-carbon, E1cB.

Chlorine can also act as an electrophilic center when connected with an electron-deficient carbon atom.

There are frequent cases where a chlorine substituent promotes more than one transformation of a molecule.

We have been working in the field of organic synthesis, mainly new methodology and search for new reactions, and in our studies, chlorine was a widely used substituent for a variety of purposes, a real workhorse. Moreover, in our research, it appears that we have used almost all possibilities provided by chlorine substituents; thus, we will exemplify a variety of applications of chlorine using mainly our work.

## 2. Nucleophilic Substitution of Chlorine

### 2.1. In Aliphatic Compounds

The nucleophilic substitution of chlorine (or bromine) in organic molecules is perhaps the most frequently used reaction in organic chemistry. A variety of nucleophiles—inorganic anions, uncharged nucleophiles (amines, phosphines, etc.), organic anions, particularly carbanions—can afford the substitution. The reaction was thoroughly studied and mechanistic questions are clarified in detail. Of particular interest is nucleophilic substitution by carbanions, commonly referred to as alkylation, because it is the general way of construction of carbon skeletons. In these reactions, the key problem is efficient and economical way of generation of carbanions.

We have discovered and introduced to the practice of the organic synthesis catalytic method of the generation of carbanions by its action on carbanion precursor-concentrated aqueous sodium hydroxide solution in the presence of catalysts: tetraalkylammonium (TAA) salts—a source of lipophilic cations [1,2,3,4,5,6,7].

The first example of the application of this catalytic system was, as elaborated by us, the synthesis and manufacturing of 2-phenylbutyronitrile via the alkylation of phenylacetonitrile with ethyl chloride in the presence of 50% aqueous NaOH and benzyltriethylammonium chloride, TEBA (Figure 1) [8,9].

Phase-transfer ethylation of phenylacetonitrile was also presented as a procedure in *Organic Syntheses* [10].

Under identical conditions, a variety of CH acids (arylacetonitriles, acidic hydrocarbons, esters, ketones, sulfones, etc.) and OH, NH acids were efficiently alkylated with alkyl chlorides and bromides. Some examples are given in Figure 2.

The PTC procedure is also applicable for the synthesis of chiral amino acids via the enantioselective alkylation of protected glycine and other amino acids, catalyzed by properly chosen chiral tetraalkylammonium salts, which, due to the secondary interactions, can differentiate the free energies of diastereoisomeric transition states, and thus assure a high induction of enantioselectivity [11,12,13,14].

Under conditions described above, the reaction is carried out in a system of two immiscible phases. The organic phase is a mix of the reagent–carbanion precursor, an alkylating agent and, when necessary, a small amount of a solvent—whereas the inorganic phase consists of the concentrated aqueous solution of sodium hydroxide. The boundary between two immiscible liquid phases A and B (see Figure 1) is in fact a region in which, due to thermal motion, components of both phases can mix to form a kind of interfacial region, in which there is gradient of the concentration of the components of phases A and B. Due to short residence time, only very fast reactions, such as deprotonation, between these components can proceed. The interfacially generated carbanions are in low concentration, and cannot migrate into the organic phase being associated with hydrated sodium cations.

When a lipophilic TAA salt is added to such a system, an ion exchange takes place in the interfacial region between the TAA salt and the carbanion, with the formation of a lipophilic ion pair: carbanion/TAA cation. This ion pair migrates from the interfacial region to the organic phase, where carbanion reacts with the chloroalkane, with the formation of the alkylation product, whereas liberated TAA salt enters the catalytic cycle (Figure 3).

This methodology, commonly known as Phase Transfer Catalysis, PTC, is the most efficient, economic and green procedure for the alkylation of carbanions. This is because under PTC conditions, the alkylation can proceed without organic solvents, when starting materials are liquid, or with minimal quantities of solvents. On the other hand, large quantities of strictly anhydrous solvents are necessary, when generation of the carbanions proceeds under the action of *t*-BuOK, NaNH_2_, NaH, etc. under classical conditions, instead of inexpensive aqueous NaOH.

Under PTC conditions, the carbanions enter the reaction as TAA salts, in which the interactions between the anion and the cation are of an electrostatic nature only. Consequently, the carbanions are highly nucleophilic and the reaction rate constants take large values. As a result, under PTC conditions, reactions proceed rapidly, despite a low carbanions concentration in the organic phase. Additionally, a low concentration of reacting carbanions that cannot exceed the concentration of the catalyst is compensated by a high concentration of the alkylating agent. Due to the low concentration of carbanions in the organic phase, even reactions conducted without an organic solvent mimics high dilution conditions. Carbanions, generated as a result of acid–base equilibrium in the interfacial region, migrate to the organic phase as lipophilic ion pairs with catalyst cations, hence leaving the equilibrium site. According to the law of mass action, this causes the equilibrium to be shifted to the right. Therefore, many organic compounds of low acidity (to the pKa value of around 24) are transformed into anions under such conditions. Taking into account the pKa of water (15.7), we can conclude that the lipophilic cations of TAA catalyst exert a hyperbasic effect on aqueous sodium hydroxide.

This mechanism was thoroughly studied, presented in a few publications [6,15] and supported experimentally [16,17,18]. Another proposed mechanism [19]—the ion exchange of TAA salt with sodium hydroxide and subsequent transfer of TAA hydroxide into the organic phase, where the deprotonation takes place, should be ruled out due to high hydration energy of hydroxide anions, and thus an unfavorable extraction equilibrium (Figure 4).

Interestingly, the PTC alkylation proceeds efficiently with alkyl chlorides and bromides, but not with alkyl iodides. This is because iodide anions inhibit the catalytic generation of carbanions [20].

For the alkylation of compounds being relatively strong CH acids, and sensitive toward hydrolysis (diethyl malonate, methyl cyanoacetate, ethyl acetoacetate, etc.), less active bases, such as solid potassium carbonate, assure good results (Figure 5) [21,22,23].

### 2.2. Substitution of Chlorine in Aromatic Rings

Chlorine connected with the π-system, as for instance in vinyl chloride or chlorobenzene, does not enter S_N_2 nucleophilic substitution. The situation is changed when the π-system become electron deficient, as for instance in *p*-chloronitrobenzene, β-chloroacrylonitrile or acyl chlorides. In such compounds, chlorine is readily replaced by nucleophiles via an addition–elimination mechanism. Of particular importance is the substitution of chlorine in activated positions of nitroarenes with a variety of nucleophiles. The reactions with carbanions proceeds satisfactorily, provided they are not in the form of tight ion pairs with alkali metal cations in nonpolar solvents. We have applied PTC conditions for the nitroarylation of carbanions of arylalkanenitriles by *o*- and *p*-chloronitroarenes. These conditions assure excellent results of substitution in methinic carbanions. Methylenic carbanions, e.g., phenylacetonitrile, upon nitroarylation form highly acidic phenyl nitroarylacetonitriles that are immediately deprotonated and exist in the reaction system in the form of ion pairs with TAA cations; thus, the catalytic process is arrested. A low nucleophilic activity of highly stabilized carbanion prevents its further reactions with chloronitrobenzene, thus bis-nitroarylation (Figure 6) [24].

## 3. β-Elimination

The base-induced β-elimination of hydrogen halide from haloalkane is an important way of synthesis of alkenes. It requires the use of a strong, non-nucleophilic base, such as a hydroxide anion, for the abstraction of proton, with the synchronous departure of the halide anion from the vicinal carbon atom (Figure 7).

The application of PTC methodology for this β-elimination is limited, because it requires the transfer of hydroxide anions into the organic phase. However, ion exchange equilibrium between the organic and aqueous phase of the hydroxide and chloride anions is unfavorable for hydroxide anions (Figure 4). This problem was solved by the use of cocatalysts–organic acids (YHs) that can be deprotonated in the interfacial region with aqueous NaOH, producing lipophilic anions Y^−^ of high basicity, but low nucleophilicity. Such anions can be transferred into the organic phase as ion pairs with TAA cations and abstract protons from haloalkanes to give products of β-elimination (Figure 8).

Using the model β-elimination reaction of hydrogen bromide from bromocyclohexane, we tested a variety of YHs and have shown that the most efficient cocatalyst is mesitol, with a reasonable OH acidity and basicity of its anion, whereas steric bulkiness prevents substitution, so this readily available compound is the most recommended cocatalyst for β-elimination (Figure 9) [25,26].

## 4. Chlorine Substituent as Promotor of CH Acidity

### 4.1. Generation and Reactions of Dihalocarbenes

Due to strong electron-withdrawing effect and presence of unoccupied 3d orbitals, chlorine connected with a carbon atom exerts a moderate carbanion-stabilizing effect, thus facilitating the abstraction of the proton. Depending on the structure of the molecule, this effect ranges from 3 to 4 pKa units.

The acidity of chloromethane is negligible, and the estimated pKa of dichloromethane is around 30; on the other hand, chloroform is relatively acidic, estimated pKa ~15. In the α-chlorocarbanions, generated via the deprotonation of methylene chloride and chloroform, a negative charge is located on chloro substituents; thus, they are very unstable and undergo rapidly α-elimination of the chloride anion to form mono- and dichlorocarbenes, respectively. The produced carbenes are very active electrophiles; they add to alkenes with the formation of mono- and *gem*-dichlorocyclopropanes, respectively. Due to facile deprotonation of chloroform and subsequent α-elimination, dichlorocarbene becomes an active agent widely used for the synthesis of cyclopropanes. The carbene is highly sensitive to water and other nucleophiles; thus, the known procedures required the use of *t*-BuOK under strictly anhydrous conditions (Figure 10) [27,28,29].

We have, however, found that dichlorocarbene (DCC) could be generated and added efficiently to alkenes by the treatment of a mixture of chloroform and alkenes with a concentrated aqueous sodium hydroxide solution in the presence of the TAA catalyst [30]. This procedure is much simpler and usually assures good results (Figure 11).

Hundreds of examples of the syntheses of dihalocyclopropanes from PTC-generated dihalocarbenes and unsaturated compounds are now described, including adducts of alkenes, unconjugated and conjugated di- and polyenes, allenes, cumulenes, and alkenes substituted with different kinds of substituents (haloalkenes; unsaturated ethers, acetals and ketals; esters of unsaturated alcohols; alkenes substituted by sulfur, nitrogen, silicon atoms; and some unsaturated carbonyl compounds and many others) [28,29,31].

There are many observations that DCC generated under PTC conditions is more efficient in reactions with partners of low activities—addition to alkenes of low nucleophilicity or insertion into the CH bond. For instance, PTC-generated DCC adds to *trans*-stilbene to form a dichlorocyclopropane derivative with a high yield (Figure 12), whereas under classical conditions, the addition practically does not proceed [32].

This substantial difference in efficiency cannot be explained by differences in activities, because the carbenes are kinetically free species and should not “remember” ways of their generation. Indeed, the study of activities of DCC via competitive experiments with various alkenes show identical activities, regardless of the way of generation [33]. Explanation of this substantial difference in effectiveness of DCC is very simple. Under the classical conditions, chloroform is treated with potassium *t*-butoxide in an anhydrous inert solvent. The produced intermediate potassium trichloromethylide is insoluble in such solvents, and its dissociation produces insoluble potassium chloride; thus, both of these processes are irreversible. Therefore, the carbene, once generated, should be consumed. When a desired reaction is not a fast process, DCC is consumed by side reactions, e.g., with *t*-butanol (Figure 10).

On the other hand, under PTC conditions, chloroform is deprotonated in the interfacial region and trichloromethyl anions enter the organic phase as soluble ion pairs with TAA cations. Further reaction: the dissociation of trichloromethyl anion is a reversible process, because all components are soluble in the reaction medium; thus, when the desired reaction of DCC is not a fast process, the carbene adds a chloride anion and is kept “ready for use” for a longer time (Figure 11 and Figure 13).

This simple explanation of the high efficiency of PTC-generated DCC is unambiguously confirmed, because it was shown that the rate of consumption of chloroform is a function of nucleophilic activity of alkenes, in other words, the rate of consumption of DCC, thus the removal of the carbene from the equilibrating system [34].

In this system, DCC has practically no possibility of meeting hydroxide anions or water, because it is generated in the organic phase; as a consequence, its hydrolysis occurs only to a very small extent. The interfacially generated trichloromethyl anion can also dissociate to form interfacially located DCC that can hydrolyze. However, the produced chloride anions accumulate at the surface of the aqueous phase and prevent hydrolysis. The continuous generation of DCC in the interfacial region, where it stays in equilibrium with trichloromethyl carbanion and interfacially located chloride anions, has also been confirmed by early observations that trialkylamines catalyze the addition of DCC to alkenes [35]. Alkenes are nucleophiles insufficiently active to add carbenes in the interfacial region, whereas trialkylamines are much stronger nucleophiles and add irreversibly interfacially located carbene with the formation of an ammonium ylide, which acts as a base in the organic phase, deprotonating chloroform. Trichloromethyl anion thus forms dissociates and generated DCC adds to alkene. Trialkyldichloromethylammonium chloride is converted into trialkylamine and chloroform (Figure 14) [36].

Besides DCC, the α-(or γ)-elimination of hydrogen halides in a PTC system permits the generation of many other carbenes, providing the appropriate α-halocarbanions are not stabilized by electron withdrawing groups. It is worthy to note that the PTC α-elimination of HCl from dibromochloromethane in the presence of alkenes leads to a mixture of all three possible *gem*-dihalocyclopropanes [34]. This result is not surprising if one keeps in mind that the dissociation of the dibromochloromethyl anion to bromochlorocarbene is truly a reversible process. If the starting haloform contains different halogen atoms, exchange reactions afford all possible dihalocyclopropanes. It is worthy to note that the treatment of dibromochloromethane with *t*-BuOK in the presence of an alkene leads to pure *gem*-bromochlorocyclopropanes, thus the irreversibility of generation of DCC (Figure 10) is confirmed.

### 4.2. Generation of α-Halocarbanions and Their Addition to Activated Carbon–Carbon Multiple Bonds

The introduction of chlorine in position α- to the functional groups of nitriles, esters, sulfones, etc., increases the CH acidity of these compounds. Thus, α-chloronitriles, esters, sulfones, etc., undergo relatively readily deprotonation to form α-chlorocarbanions. These carbanions are unable to enter α-elimination, because the negative charge is located mostly on the functional groups, but are unstable for other reasons. Nevertheless, they are sufficiently stable to react with such active π-electrophilic partners as aldehydes, ketones and Michael acceptors. In these adducts, the chlorine substituent is situated in position γ to the nucleophilic center; thus, the next step is fast intramolecular 1,3-substitution to form oxiranes or cyclopropanes, respectively. These reactions, known as the Darzens reaction and the Michael reaction, are widely used in organic synthesis.

We showed that because of the presence of chlorine substituents α- to the functional group of aliphalic nitriles, esters, sulfones, etc., they become sufficiently strong CH acids; hence, the generation of α-chlorocarbanions can be effected under the action of the concentrated aqueous NaOH solution in the presence of TAA catalysts. Therefore, a simple PTC methodology can be applied for the synthesis of a variety of oxiranes [37,38,39,40] and cyclopropanes (Figure 15) [41,42,43,44].

It was subsequently shown that the use of properly chosen chiral TAA salts as the catalysts in these reactions allows the enantioselective synthesis of oxiranes and cyclopropanes [12].

### 4.3. Reactions of α-Chlorocarbanions with Electron-Deficient Arenes

An interesting general reaction that we discovered was between α-chlorocarbanions and electron-deficient arenes, particularly nitroarenes. For instance, the carbanion of chloromethyl phenyl sulfone adds to nitrobenzene at positions *o*- and *p*- to form adducts (σ^H^ adducts). These adducts are short-lived species; nevertheless, they undergo the base-induced β-elimination of HCl at the expense of the ring hydrogen to form anionic products, that upon protonation gave a mixture of *o*- and *p*-nitrobenzyl phenyl sulfones. Interestingly, the carbanion of chloromethyl phenyl sulfone adds to *p*-chloronitrobenzene at the position *ortho* to the nitro group, giving σ^H^ adducts, that upon β-elimination and protonation gave 2-nitro-5-chlorobenzyl phenyl sulfone. Substitution of the ring chlorine did not proceed (Figure 16) [45].

This unprecedented pathway of conversion of the adducts of α-chlorocarbanions to nitroaromatic rings is of general character. For this new reaction, we proposed the name the Vicarious Nucleophilic Substitution of hydrogen, VNS [46]. Further examples of this reaction are shown in scheme (Figure 17) [47,48,49,50].

Interestingly, carbanions without a chlorine substituent react with *o*- and *p*-chloronitrobenzene via the replacement of chlorine, whereas α-chlorocarbanions replace hydrogen *ortho* to the nitro group. Similarly, reactions of α-chlorocarbanions with *o*-chloronitrobenzene result in the substitution of hydrogen *para* into the nitro group. These seemingly controversial unusual observations suggest that the addition of carbanions at positions *ortho* and *para* of nitrobenzene is a fast and reversible process. Initially formed σ^H^ adducts of α-chlorocarbanions to *o*- and *p*-chloronitrobenzene undergo fast further conversion via the base-induced β-elimination of HCl. On the other hand, σ^H^ adducts of carbanions, which do not have such a possibility, dissociate, and further, the slower addition of the carbanions at the position occupied by chlorine results in nucleophilic substitutions of chlorine. Thus, it was unambiguously shown that the initial fast addition of carbanions to *o*- and *p*-chloronitrobenzenes proceeds at positions occupied by hydrogen. Therefore, the known and generally accepted mechanism of the S_N_Ar reaction should be corrected.

A peculiar effect of chlorine in the nitroaromatic rings should be mentioned: it activates positions occupied by hydrogen towards nucleophilic addition (e.g., the VNS in positions *ortho* of *p*-chloronitrobenzene is 120 times faster than in nitrobenzene) and protects positions it occupies against nucleophilic addition [51].

Further thorough mechanistic studies resulted in the formulation of a corrected, general mechanism of aromatic nucleophilic substitution [52,53,54,55].

Interestingly, although reactions of α-chlorocarbanions with a variety of aliphatic π-electrophiles have been widely applied, only 1,3-intramolecular substitution to form three-membered rings was reported as a further conversion. The possibility of the base-induced β-elimination was not mentioned.

On the other hand, we have shown that, in some cases, under proper conditions, adducts of α-chlorocarbanions to Michael acceptors can react further, not along the 1,3-intramolecular pathway, but undergo the base-induced β-elimination of HCl. Thus, the VNS reaction can proceed also in aliphatic π-electrophiles (Figure 18 and Figure 19) [56,57,58].

The same peculiar effect of chlorine, as in aromatic nitro compounds, was observed in nucleophilic reactions of aliphatic π-electrophiles [59]. For instance, α-chlorocarbanion replaces the 3-hydrogen atom in 2-chloronaphthoquinone faster than in naphthoquinone itself, but the replacement of 3-chlorine in 2,3-dichloronaphthoquinone is much slower (Figure 20).

### 4.4. γ-Halocarbanions

Being engaged in the chemistry of α-chlorocarbanions, we were looking for unprecedented reactions of β-, γ- and δ-chlorocarbanions. The thorough analysis of β-chlorocarbanions indicate that the elimination of the chloride anion to form alkenes, the E1cB elimination of HCl, is a very fast process; thus, the only observed intramolecular process is protonation, which can be monitored as isotope exchange. 

We were, however, convinced that, although the only known conversion of γ-chlorocarbanions is intramolecular 1,3-nucleophilic substitution to form cyclopropanes, there is the possibility of finding an intermolecular reaction of γ-chlorocarbanions.

There are two main methods of the generation of γ-chlorocarbanions: the deprotonation of appropriate precursors—CH acids and addition of α-chlorocarbanions to Michael acceptors. Precursors of γ-chlorocarbanions can be prepared in advance or generated in situ, as shown in Figure 21.

The frequently used second pathway of the generation of γ-chlorocarbanions is the generation of α-chlorocarbanions in the presence of Michael acceptors.

Irrespective of the pathway of generation, the γ-chlorocarbanions subsequently enter fast 1,3-intramolecular nucleophilic substitution to form substituted cyclopropanes.

We expected that the generation of γ-chlorocarbanions in the presence of an active electrophile, such as benzaldehyde, should result in addition to the carbonyl group faster than intramolecular substitution. Subsequent 1,5-intramolecular substitution by the negatively charged oxygen of the produced aldol type adduct should give substituted tetrahydrofurans. Indeed, treatment of a mixture of 4-chlorobutyronitrile and benzaldehyde in THF with *t*-BuOK gave 2-phenyl-3-cyanotetrahydrofuran (Figure 22) [60]. It was the first observation of the intermolecular reaction of a γ-chlorocarbanions. This observation is within the general character of γ-chlorocarbanions and active electrophiles, e.g., Michael acceptors [61] and imines [62]. Thus, this new process is a general pathway of synthesis of 5-membered rings: substituted tetrahydrofurans, cyclopentanes and pyrrolidines [60,61,62].

Following this line, we also showed that treatment of a mixture of ethylene chlorohydrine and benzaldehyde with a base gave ethylene acetal of benzaldehyde (Figure 23) [63].

We also found that chlorine, in 4-chlorobutyronitrile and 3-chloropropylphenylsulfone, exerts a moderate carbanion-stabilizing effect on the α-position to the functional group (Figure 24) [64].

Thus, the well-known observation that the alkylation of methylenic carbanions with ethylene chloride and bromide always gives cyclopropanes can be rationalized. The reaction cannot be arrested upon the introduction of a 2-haloethyl group, because initial products of alkylation become stronger CH acids and are immediately deprotonated and enter fast intramolecular 1,3-substitution.

It is worthy to note that the addition of carbanions of *t*-butyl chloroacetate or aryl halomethyl sulfone to aldehydes, carried out in the presence of LDA/THF at −70 °C, yields corresponding halohydrines in good yields. Treatment of the halohydrine with methanesulfonyl chloride and trimethylamine converts it into substituted haloalkenes with high *E*/*Z* selectivity (Figure 25) [65].

## 5. Chlorine as Electrophilic Center

Chlorine connected with an electron-withdrawing carbon center exhibits an electrophilic character; thus, it can react with nucleophiles, particularly carbanions, affording electrophilic chlorination. The most readily available sources of electrophilic chlorine are CCl_4_ and hexachloroethane (Figure 26).

We studied the synthetic applicability of such electrophilic chlorinating agents in reactions with carbanions. The reaction of the carbanion of phenylacetonitrile with CCl_4_ under PTC conditions resulted in the formation of dicyanostilbene [66].

The initial step of this process is undoubtedly chlorination of the carbanion, followed by further transformation of the chloronitrile.

The initial formation of the chloronitrile was confirmed by tri-component reactions—phenylacetonitrile, benzaldehyde or acrylonitrile—and CCl_4_ under PTC conditions.

The carbanion of phenylacetonitrile in the organic phase can react with benzaldehyde and CCl_4_. The fast addition of the carbanion to benzaldehyde that resulted in formation of an aldol-type anion is a reversible process and does not result in the formation of a stable final product. On the other hand, the reaction of this carbanion with CCl_4_ proceeds irreversibly and gives α-chlorophenylacetonitrile which is a stronger CH acid than phenylacetonitrile; thus, it is immediately deprotonated. Addition of the generated α-chlorocarbanion to benzaldehyde is followed by rapid intramolecular substitution in the irreversible process to form, as the final product, oxirane (the Darzens reaction) (Figure 27). 

The high yield of the final product 1,2-diphenylcyanooxirane (80%) indicates that all steps of this multistep process proceed with high selectivity. This process is of a general character with respect to aromatic aldehydes, and is possible because of the specific features of PTC conditions. A similar sequence of reactions proceeda between phenylacetonitrile and the Michael acceptor, acrylonitrile and CCl_4_ under PTC conditions giving 1,2-dicyanophenylcyclopropane (Figure 27) [67,68].

Phenylchloroacetonitrile is not readily available; therefore, this method of generation in situ of its carbanion for the reactions with aldehydes and Michael acceptors is of practical value.

Interestingly, in the reaction with the enolate of aromatic ketones under PTC conditions, CCl_4_ serves as a source of electrophilic chlorine and nucleophilic trichloromethyl carbanion (Figure 28) [68].

In the first step, chlorination of the enolate results in the formation of α-chloroketone and trichloromethyl carbanion. This carbanion, generated in the vicinity of the ketone, adds to the carbonyl group. Subsequent intramolecular substitution gives an oxirane-containing trichloromethyl group.

Particularly, interesting results were obtained in the reactions of CCl_4_ with sulfone-stabilized carbanions, because α-chlorosulfonyl carbanions are relatively stable entities. For instance, benzyl phenyl sulfone under PTC conditions undergo dichlorination. The reaction cannot be stopped on the stage of monochlorination, because initially formed α-chlorocarbanion is a stronger CH acid; thus, is immediately deprotonated to form α-chlorocarbanion that undergoes further chlorination. The dichlorinated sulfone is an active source of electrophilic chlorine; thus, these reactions are a reversible processes. Nevertheless, these in situ-generated α-chlorocarbanions can be trapped by other electrophilic partners. We used this sequence of reactions for the VNS reaction. Thus, PTC-generated carbanions of phenyl and neopentyl benzyl sulfonates treated with CCl_4_ gave dichlorinated products. The reaction of the equimolar mixture of the starting and dichlorinated esters with nitroarenes and excess of KOH gave products of VNS in good yields (Figure 29) [69].

This concept was also applied for the reaction of carbanions of allylic sulfones [70]. In order to avoid the generation of dichlorocarbene that is produced during chlorination by CCl_4_, which can add to the double bond, hexachloroethane instead of CCl_4_ was used as the chlorination agent. The PTC reaction of dimethylallyl phenyl sulfone with hexachloroethane gave monochlorinated sulfone, a stronger CH acid; thus, it is immediately deprotonated. The produced α-chlorocarbanion is further chlorinated to give dichlorosulfone. However, when in the system and there are other active electrophiles—benzaldehyde or acrylonitrile—the α-chlorocarbanion enters a fast reaction with these electrophiles to give oxirane or cyclopropane (Figure 30) [70].

In our opinion, the chlorination of carbanions proceeds via the direct nucleophilic attack on an electron-deficient chlorine-halophilic reaction. There was, however, the concept that the reaction proceeds via initial single electron transfer (SET) from the carbanions to CCl_4_ to form a radical anion-radical pair (RARP) (Figure 31) [71].

Subsequently, the radical abstracts chlorine atom to form chlorinated products, or RARP dissociates. On the basis of experiments with a fast radical clock, we showed that the chlorination does not proceed via RARP (Figure 32) [72].

Another valuable reaction of CCl_4_ as a source of electrophilic chlorine proceeds with phosphorous nucleophiles (triphenylphosphine, dimethyl phosphite, etc.), for instance Appel [73] or the Atherton–Todd reaction [74,75].

We have presented rich possibilities connected with the chlorination of carbanions by CCl_4_ in detail because, in our opinion, they are insufficiently recognized.

## 6. Conclusions

There are thousands of examples of reactions that proceed because of the presence of chlorine substituents in organic molecules. We have chosen the most convincing reactions, mostly designed by us. 

We believe that the presentation in this paper of a variety of reactions induced by chlorine substituents in organic molecules justifies the conclusion that they are the most universal and versatile workhorses in organic synthesis.

It is worthy to note that PTC is a particularly convenient methodology of performing most of these reactions, because of the moderate energy of hydration of chloride anions.

## Data Availability

Data sharing not applicable.

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
