# Peer review of "Chlorine in an Organic Molecule, a Universal Promoter—Workhorse—Of Reactions"

_molecules, 2023, doi:10.3390/molecules28247957_

Round 1

Reviewer 1 Report

Comments and Suggestions for Authors

Comments: The review article entitled “Chlorine in an organic molecule, a universal promoter – work horse – of reactions” is overall a good scientific effort made by MieczysÅ‚aw MÄ…kosza and MichaÅ‚ FedoryÅ„ski. The background and purpose of this study is interesting and but the review article is not properly handled by the authors. I believe that this article fulfills the quality to be published in Journal of molecules and can be of interest for its readership after the suggested modifications.

Recommendation: Accept with major revision

1.      The title need to be modified to make it more appealing. Also, the tile is about the choline synthetic utility but in some example there is bromine too (line 206).

2.      Remove the word “Thanks” from the whole article which has been used five times. A more suitable word can be used.

3.      Abstract need be precise and should not be in the form of multiple paragraphs. Also, abstract is simply the reused version of paragraph 3 of the introduction.

4.      Line  14 and 87 : “via”  can be written in italic form

5.      For the Keywords, the authors should fallow the journal format as well as reduce the number of words allowed by the journal of molecules.

6.      The introduction part (1) needs to be re-write as in the current form there is no continuity as well as there is no references (even a single reference has not been cited). The authors need to include suitable references in this part.

7.      Also, lines 27-72: the introduction part, the authors need to modify it and prolong it properly in order to link their study/research with the literature to make the background of their study more authentic.

8.      Line 56:  in the introduction part (SRN1) complete abbreviation

9.      All the structures should be re-drawn using chemdraw style 1996, the structures should be clean, angle and stereochemistry of the structures should be adjusted according to the stander. Majority of the Sp3 hybridized carbons; the authors have drawn the bond angle as 90o which need to be 109.5o by cleaning the structure these angles will be adjusted to the standard.

10.  Figure 1 the quality of the structure should be improved and then centralized.  

11.  Line 65 should be in concluding section of the introduction.

12.  Line 86, “The first example of applications of this catalytic system was elaborated by us syn- thesis” should be reviewed.

13.  Figure 1, labelling of graph on the left side is missing.

14.  Conclusion should be revised as well as extended, it should be under discussion why chlorine substituents are effective reaction reagent in PCT synthesis and what are the future prospects. Currently, the conclusion is only 2 and half lines which is not enough.

15.  The fallowing article need to be cited: Org. Biomol. Chem., 2019, 17, 519 (DOI: 10.1039/c8ob02110d)

16. All the references are not formatted according to the journal format. Write  DOI number for all the references.

Comments on the Quality of English Language

Minor editing of English language is needed

Author Response

We are very grateful for the deep analysis of our paper by the reviewer and in almost all cases follow their recommendations in preparing the revised version.

1. We consider the title adequate to our intent and contents of the paper. We have added one sentence indicating that bromine can play similar action, promote similar reactions, however lower energy of bromine-carbon bond, larger size of the atom, and electronic configuration, make it less universal.

2. We follow this recommendation.

3. We have abbreviated the abstract.

4. Introduced.

5. The journal allows ten keywords, we have 9.

6 and 7. We do not understand this suggestion. Since there are thousands of reactions promoted by chlorine atom present in the molecules that are presented in textbooks, we consider that references are not necessary.

8. We added a few words that fulfill the reviewer's request: "[...] transfer to chloroarenes followed by dissociation of the produced anion radical".

9. Structures on schemes 4, 6, 7, 14, 15, 18, 19, 25, 26, 27 were corrected.

10 and 13. Figure 1 was corrected.

11. In our opinion this sentence is in a proper place.

12. Corrected. The sentence is now unambigous.

14. Conclusion was revised.

15. The paper suggested by reviewer is in our opinion beyond the scope of our article, in which reactions of chlorine containing compounds are discussed, not methods of their preparation.

16. It seems all the references are formatted according to the journal format. We are unable to write DOI numbers.

Reviewer 2 Report

Comments and Suggestions for Authors

The manuscript requires major revision on the basis of following comments.

1.       Introduction: Line39-40 are same as that of abstract part. Please change these lines.

2.       Introduction contains no reference which makes the manuscript very weak. Please cite the relevant articles.

3.       Major improvement is required in introduction section. The authors are requested to please take a literature survey. Lot of groups have worked on the compounds containing chlorine atoms in past.

4.       Nucleophilic substitution of chlorine (or bromine) in organic molecules is perhaps

the most frequently used reaction in organic chemistry. How? Please justify.

5.       The manuscript is purely based on the synthesis. The authors are requested to perform some characterizations (single crystal XRD, FT-IR, UV-Visible, NMR) to conform that the reactions that they are proposing can give fruitful results. Or at least perform single crystal XRD of few compounds.

6.       Improve conclusion section. Please add 3 to 4 most interesting findings in it.  

Comments on the Quality of English Language

Some grammatical errors are observed. Please read whole manuscript carefully.

Author Response

We are very grateful for the deep analysis of our paper by the reviewer and in almost all cases follow their recommendation in preparing the revised version.

1. We have abbreviated the abstract.

2. We don't understand this suggestion. Since there are thousands of reactions promoted by chlorine present in the molecules that are presented in many textbooks, we consider that references are not necessary.

3. Yes, we suppose that majority of organic chemists used and use a variety of reactions promoted by presence of chlorine in organic molecules.

4. Nucleophilic substitution of chlorine or bromine in the reactions with inorganic and organic anions, including carbanions, is in the program of undergraduate textbooks and laboratory courses.

5. Yes, it is our intention to discuss exclusively reactions of interest in organic synthesis. In the selected examples products of the reactions described are unambigously identified and characterized by the mentioned spectral and physicochemical techniques. Describing this is beyond the scope of this paper.

6. It is done.

Reviewer 3 Report

Comments and Suggestions for Authors

An interesting collection of syntheses in which the presence of chlorine atoms affects the course of the reaction. Prof. MÄ…kosza and coworkers discovered and developed many of the reactions presented, especially phase transfer catalysis, vicarious nucleophilic substitution, formation and reactions of carbenes, and many others.
For better readability of the entire manuscript, please standardize the reaction schemes, especially when presenting charges. A negative charge is usually represented as a minus sign, but the minus is circled in Schemes 4, 7, and 26. In Scheme 7, please correct the third arrow. It must come from the center of the bond, and the arrowhead must be outside the chlorine atom to show that the chlorine leaves as the chloride anion.

Comments on the Quality of English Language

Please improve your grammar, especially punctuation. Many commas must be added, especially after introductory words and phrases. An even bigger problem is articles a, an, and the. Many noun determiners need to be included.

Some examples:

Line 35: Therefore, we

Line 39: the electronic

Line 39 the charge

Line 39: the nucleus,

Line 42: a chloride anion

Line 42: the electron-withdrawing

Line 68: and in our studies, chlorine

Line 69: Moreover, in our research

Line 70: chlorine substituents; thus, we

Line 80: In these reactions, the key problem is an efficient and

And many more!!!

Author Response

We are very grateful for the deep analysis and kind words about our paper. We followed all recommendations in preparing the revised version (suggestion concerning improvement of quality of the structure presented on Scheme 7, signs on Schemes 4, 7 and 26).

Round 2

Reviewer 1 Report

Comments and Suggestions for Authors

I have already made the comments of version-1 but the authors have not addressed those comments, suggestions, and modifications. Still those comments need to be addressed. Also a point by point answers to all the questions is required.  

The authors need to improve the article by incorporating all the suggested modifications. In the current state, this article does not fulfil the standard of molecules. The authors have modified the conclusion as "There are thousands of examples of reactions that proceed because of the presence of chlorine substituents in organic molecules. We have chosen the most convincing reactions, mostly designed by us." so too much self citations i.e out of 75 references, there are 48 self cited references of only one author. The authors need to cite other relevant research too.   

Reviewer 2 Report

Comments and Suggestions for Authors

The authors have improved manuscript in response to all the corrections and suggestions. I am recommending acceptance of the manuscript in its present form.